# Determining the Levels of Cortisol, Testosterone, Lactic Acid and Anaerobic Performance in Athletes Using Various Forms of Coffee

**DOI:** 10.3390/nu16193228

**Published:** 2024-09-24

**Authors:** Melda Gür, Vedat Çınar, Taner Akbulut, Kenan Bozbay, Polat Yücedal, Mehdi Aslan, Gökçe Avcu, Johnny Padulo, Luca Russo, Joanna Rog, Gian Mario Migliaccio

**Affiliations:** 1Institute of Health Sciences, Faculty Sport Science, Fırat University, Elazig 23119, Turkey; dytmeldaagur@gmail.com (M.G.); kenanbozbay21@gmail.com (K.B.); gokceavcu@hotmail.com (G.A.); 2Department of Physical Education and Sport, Faculty Sport Science, Fırat University, Elazig 23119, Turkey; cinarvedat@hotmail.com; 3Department of Coaching Education, Faculty Sport Science, Fırat University, Elazig 23119, Turkey; takbulut@firat.edu.tr; 4Department of Coaching Education, Faculty Sport Science, Munzur University, Tunceli 62100, Turkey; yucedalpolat@gmail.com; 5School of Physical Education and Sports, Siirt University, Siirt 56100, Turkey; mhdiasln@hotmail.com; 6Department of Biomedical Sciences for Health, Università degli Studi di Milano, 20133 Milan, Italy; johnny.padulo@unimi.it; 7eCampus University, 22060 Novedrate, Italy; 8Laboratory of Human Metabolism Research, Department of Dietetics, Institute of Human Nutrition Sciences, Warsaw University of Life Sciences, 02-787 Warsaw, Poland; rog.joann@gmail.com; 9Department of Human Sciences and Promotion of the Quality of Life, San Raffaele Rome Open University, 00166 Rome, Italy; gianmario.migliaccio@uniroma5.it; 10Maxima Performa, Athlete Physiology, Psychology, and Nutrition Unit, 20126 Milano, Italy

**Keywords:** biochemical markers, coffee, caffeine, exercise physiology, physical performance, perform aid, hormone, metabolism, wingate test

## Abstract

Background: Coffee is considered one of the most preferred and consumed beverage types in the world, and caffeine is known to increase physical performance due to its ergogenic properties. The aim of this study is to examine the effects of coffee consumption in different forms on cortisol, testosterone, lactic acid and anaerobic performance levels. Methods: A total of 15 licensed male football players participated in the research voluntarily. The research was implemented in a single-blind, counterbalanced, randomized and crossover study design. Participants were given caffeinated coffee (CK), decaffeinated coffee (placebo) (DK), powdered caffeine (in a gelatin capsule) (PC) and powdered placebo (maltodextrin in a capsule) (PM) on different days, and the Wingate test protocol was performed after the warm-up protocol. Blood samples were collected post-test. Cortisol, testosterone and lactic acid levels in the serum samples taken were determined by the ELISA method. Results: As a result, it was revealed that caffeinated coffee given to participants who exercise increased anaerobic power. However, it was observed that lactic acid levels were higher in placebo and decaffeinated coffee. The highest level of cortisol was found in caffeinated coffee and powdered caffeine compared to the placebo. Testosterone values were observed to be highest in caffeinated coffee and decaffeinated coffee compared to a placebo. Conclusions: The study suggests that the type of caffeine is a factor that affects absorption rate, which impacts performance and hormone levels.

## 1. Introduction

Enhancing athletic performance is critical for athletic success, and achieving this success requires a multidisciplinary approach, including physiology, psychology, biomechanics, nutrition, exercise science and mental preparation [1,2,3,4,5]. One of the most important factors influencing athletic success is nutrition. Athletes use nutritional ergogenic aids to increase endurance, achieve targeted performance, facilitate rapid recovery and speed up post-exercise recovery [6,7,8]. One of the most used ergogenic aids for this purpose is caffeine [9]. Caffeine is a natural alkaloid found in coffee, tea, cocoa and energy drinks and has significant physiological effects on performance [10]. Caffeine acts as a stimulant on the central nervous system, reducing the feeling of fatigue and increasing alertness. It also improves anaerobic and aerobic performance by increasing muscle contraction strength and endurance. By promoting the mobilization of fatty acids, it supports energy production and delays glycogen use, thereby enhancing endurance performance [10,11,12,13].

Another effect of caffeine is its ability to enhance performance by increasing endogenous testosterone secretion. Testosterone is a crucial hormone that enhances strength and endurance, and the increase in these levels through caffeine consumption results in a marked improvement in sports performance [14]. In addition to its performance-enhancing effects, caffeine influences several key hormones that play a critical role in exercise. Studies show that caffeine can increase testosterone levels, which is directly linked to improvements in strength and endurance, contributing to enhanced athletic performance. Caffeine also stimulates cortisol secretion, particularly at higher doses, aiding in energy mobilization during stress and exercise. These hormonal changes are essential as they help regulate energy availability, muscle repair and overall performance, highlighting the complex interaction between caffeine, hormone levels and athletic outcomes [14]. Additionally, caffeine can increase cortisol secretion when consumed in high doses or under stress, thereby facilitating energy mobilization. Cortisol is a hormone that regulates the body’s stress response and plays a significant role during exercise. This hormone affects the body’s energy production and use, modulates the immune response and stimulates the central nervous system to enhance endurance, strength and power output. These effects are achieved by blocking adenosine receptors, which increase wakefulness and reduce fatigue [14]. These effects of caffeine highlight the critical roles of cortisol, testosterone and lactic acid in sports performance.

However, the effects of caffeine can vary from person to person due to variables such as genetic factors and metabolic rate [15]. Another important factor in the variability of caffeine’s effects is the form in which it is consumed. Caffeine can be found in many forms, such as gels, bars, gums, lozenges, capsules or energy drinks. These different forms vary in absorption levels, volume and pH, which can affect how quickly caffeine is absorbed into the bloodstream from the buccal mucosa and intestines [16]. Therefore, many studies have examined how different forms of caffeine affect the rate at which caffeine enters the bloodstream compared to traditional tablet or coffee consumption, whether they stimulate direct connections between caffeine receptors and the brain via the oral and nasal cavities and whether they are ergogenic in training and competition scenarios [17,18,19,20]. However, these studies have examined caffeine forms separately. The present study is significant as it comprehensively compares the effects of consuming different types of caffeine on cortisol and testosterone levels and anaerobic performance in athletes. Accordingly, the research hypothesized that “different types of coffee (caffeinated coffee, decaffeinated coffee, powdered caffeine, powdered placebo) would affect lactic acid, cortisol, testosterone levels, and anaerobic performance in athletes at varying levels.” Therefore, the study aims to examine the effects of different forms of coffee on anaerobic performance and related biochemical parameters.

## 2. Materials and Methods

### 2.1. Experimental Design and Dosing Procedures

The study was a single-blind, counterbalanced, randomized, and crossover trial conducted with soccer players. Prior to the study, a G*Power analysis was conducted to calculate the statistical power and determine the required minimum sample size. According to the analysis, the minimum sample size required to detect a significant difference was determined to be at least 12 participants. The Type I error rate (alpha) was set at 0.05, the sample size was 15, and the effect size was 0.40. Under these parameters, the power of the test (1-beta) was calculated to be 0.8. [21] The study group consisted of a total of 15 volunteer male soccer players who play in the same league, with an average age of 22.50 ± 1.94 years, stature of 180.33 ± 7.12 cm, body mass of 75.20 ± 10.40 kg and body mass index of 23.03 ± 2.24 kg·m^−2^. All procedures were explained to the participants, and written consent was obtained. Prior to the commencement of the research, ethical approval was obtained from the Non-Interventional Research Ethics Committee of Fırat University with the session number 2023/14-34 dated 14 December 2023. Additionally, the study was conducted in accordance with the Declaration of Helsinki.

Inclusion criteria included healthy male soccer players without chronic diseases, non-smokers, non-alcohol users, those not taking any medication or supplements and those without a high caffeine sensitivity. Exclusion criteria included those with allergies to caffeine or other substances, those who had undergone pharmacological treatment within the last three months, those with a history of chronic diseases or serious injuries, those with psychiatric disorder histories, those using additional ergogenic aids, smokers, alcohol consumers or those following extreme diets. Participants abstained from coffee, caffeine, alcohol and exercise three days before and during the study. Dietary records were kept 72 h before the first trial, and tests were conducted at 72-h intervals, always at the same time each day. An information form created by the researchers was used to assess the inclusion criteria. The single-blind design was implemented to ensure that participants were unaware of the specific type of supplement they consumed during each trial (whether caffeinated coffee, decaffeinated coffee, powdered caffeine or placebo). However, the researchers who administered the interventions knew the content of each supplement. This method was crucial in eliminating potential psychological biases, such as expectancy effects, that could alter participants’ physiological and performance responses. By keeping participants blind to the intervention, the study aimed to measure the true effects of caffeine versus placebo, without interference from subjective assumptions about performance outcomes. Additionally, this approach was paired with a counterbalanced and randomized crossover design, where each participant received all types of supplements in different sequences, allowing them to serve as their own control. This combination of blinding and design rigor enhances the study’s internal validity, ensuring the reliability and accuracy of the results.

The study used four different forms of coffee: caffeinated coffee (CK), decaffeinated coffee (placebo) (DK), powdered caffeine (in gelatin capsules) (PC) and powdered placebo (maltodextrin in capsules) (PM). The coffee forms and dosage amounts used in the study were adjusted according to each participant’s weight. The caffeine dosage was set at 3 mg/kg, and to achieve this dosage, participants consumed 0.093 g of coffee per kilogram [22]. The dosage of coffee was calculated to ensure a caffeine intake of 3 mg per kilogram of body weight, which is a standard dose in ergogenic research. Based on the caffeine content of the coffee used in the study, we determined that 0.093 g of coffee per kilogram of body weight would provide the required caffeine dose. The calculation was performed using the formula: Coffee (g) = Target caffeine dose (mg)/Caffeine content per gram (mg/g)

An equivalent amount of decaffeinated instant coffee was used for the decaffeinated coffee (placebo). Powdered caffeine was administered in gelatin capsules at a dose of 3 mg per kilogram. Maltodextrin in the same amount was used as the powdered placebo. This method ensured that each participant received an appropriate amount of caffeine based on their body weight and allowed for consistent evaluation of the effects. No supplements other than the types of caffeine provided within the study were used. The supplements administered to participants were carefully controlled according to the study protocol, and all necessary legal and ethical approvals were obtained. The use of any additional supplements was strictly prohibited during the study, aiming to ensure the accuracy and reliability of the results.

On each test day, the Wingate Anaerobic Test was performed 60 min after the participants consumed the experimental supplements [23], followed by blood sample collection (Figure 1).

### 2.2. Taking and Analysis of Blood Samples

Fasting blood samples were collected from the forearm venous vein into EDTA tubes immediately after the test protocol on each test day between 8 and 10 a.m. The collected blood samples were centrifuged at 5000 rpm for 5 min, and the serum samples were transferred into Eppendorf tubes and stored at −80 °C until the day of analysis. Cortisol, testosterone and lactic acid levels in the obtained samples were analyzed using the enzyme-linked immunosorbent assay (ELISA) method according to the working instructions provided in the kit catalog of Reed Biotech Ltd., Wuhan, China. Absorbance readings were taken using a ChroMate Microplate Reader P4300 (Awareness Technology Instruments, Palm City, FL, USA). Plate washings were performed using an Auto-washer Bio-Tek ELX50 (BioTek Instruments, Santa Clara, CA, USA). The results for testosterone and cortisol were expressed in ng/mL, while lactic acid was expressed in mmol/L. Cortisol (Catalog Number; RE10109): The kit’s measurement range is 0.31–20 ng/mL, with a sensitivity of 0.19 ng/mL. Testosterone (Catalog Number; RE10175): The kit’s measurement range is 0.16–10 ng/mL, with a sensitivity of 0.19 ng/mL. Lactic acid (Catalog Number; RE510483): The kit’s measurement range is 0.5–10 mmol/L, with a sensitivity of 0.1 mmol/L.

### 2.3. Wingate Anaerobic Test Protocol

On each test day, prior to the tests, the athletes’ height and body weight were recorded. Participants underwent a standardized 15 min warm-up process that included a 10 min warm-up run and 5 min of stretching exercises on each test day. To assess anaerobic performance, the Wingate Anaerobic Power and Capacity Test was conducted using a Monark 834E (MONARK, Vansbro, Sweden) cycle-ergometer. The bicycle was adjusted for each participant before the test, and necessary measurements were taken during the test. Participants were verbally encouraged to maintain their performance throughout the test, which was concluded once the time was up. Anaerobic power (peak power) and capacity (mean power) values were calculated and recorded in Watts and Watts/kg using the Monark Anaerobic Test Software version 3 [24].

### 2.4. Determination of Body Weight and BMI

The participants’ height measurements were taken using a SECA 213 stadiometer, which provides high accuracy. To determine body weight and BMI (Body Mass Index) values, the AVIS 333 Body Analyzer (Jawon Medical, Co, Seoul, Republic of Korea) was used.

### 2.5. Statistical Analysis

For statistical analyses, SPSS 22 (IBM Corp, Armonk, NY, USA) and GraphPad Prism (7-day trial version) were utilized. The Shapiro–Wilk test was conducted to assess the normality of the data distribution, confirming that the data followed a normal distribution. The effects of various supplements administered at different times were analyzed and compared using Repeated Measures ANOVA. Effect sizes were calculated using eta-squared (*η*_*p*_^2^) values and interpreted based on Cohen’s criteria. Effect sizes were classified as small (<0.4), medium (0.41–0.70) and large (>0.70) [25]. The *p*-value < 0.05 was considered statistically significant.

## 3. Results

When examining Figure 2, significant differences in cortisol levels (Figure 2A) were observed among the different forms of coffee (F = 17.773, *p* < 0.05). The effect size for the cortisol variable, which demonstrated statistically significant differences, was classified as “medium” (*η*_*p*_^2^ = 0.55).

Analysis of cortisol levels revealed no significant differences among the DK, CK and PC forms. However, the PM form differed significantly from all other caffeine forms (*p* < 0.05). Figure 2B illustrates significant differences in lactic acid levels among the various forms of coffee (F = 58.137, *p* < 0.05). The effect size for the lactic acid variable, which exhibited statistically significant differences, was categorized as “large” (*η*_*p*_^2^ = 0.80). Upon evaluating lactic acid levels, no significant differences were found between the DK and PC forms. Nonetheless, the CK and PM forms showed significant differences compared to all other caffeine forms (*p* < 0.05). Finally, as depicted in Figure 2C, significant differences in testosterone levels were identified among the different forms of coffee (F = 22.780, *p* < 0.05). The effect size for the testosterone variable, which showed statistically significant differences, was deemed “large” (*η*_*p*_^2^ = 0.61). Analysis of testosterone levels indicated significant differences among all forms (*p* < 0.05).

Upon examining Figure 3A, significant differences in Maximum Power (max Watt) values were identified among the different coffee forms (F = 18.459, *p* < 0.05). The effect size for the maximum power variable, which showed statistically significant differences, was evaluated as “medium” (*η*_*p*_^2^ = 0.54). Comparisons among the DK, CK, PC and PM forms revealed that the CK form achieved significantly higher Maximum Power values compared to the other forms. The statistically significant differences among the caffeine forms are as follows: the CK form is significantly different from the DK, PC and PM forms (*p* < 0.05). Additionally, the DK form exhibited significantly higher Maximum Power values than the PM form, while the PC form also showed significantly higher values compared to the PM form. In Figure 3B, significant differences in Maximum Power (Watt/kg) values were also found among the different coffee forms (F = 24.133, *p* < 0.05). The effect size for the maximum power (Watt/kg) variable, which showed statistically significant differences, was evaluated as “small” (*η*_*p*_^2^ = 0.36). Comparisons among the DK, CK, PC and PM forms indicated that the CK form reached significantly higher values compared to the other forms. The statistically significant differences among the forms are as follows: the CK form is significantly different from the DK, PC and PM forms (*p* < 0.05). Additionally, the PC form exhibited significantly higher Max Power/kg values than the PM form, while the DK form also showed significantly higher values compared to the PM form. These results indicate that caffeinated coffee consumption has positive and significant effects on power output adjusted for body weight.

Examining Figure 3C, significant differences in Average Power values were found among the different coffee forms (F = 21.818, *p* < 0.05). The effect size for the average power variable, which showed statistically significant differences, was evaluated as “medium” (*η*_*p*_^2^ = 0.44). Comparisons among the DK, CK, PC and PM forms revealed that the CK form achieved significantly higher Average Power values compared to the other forms. The statistically significant differences among the forms are as follows: the CK form is significantly different from the DK, PC and PM forms (*p* < 0.05). Additionally, the DK form exhibited significantly higher Average Power values than the PM form, while the PC form also showed significantly higher values compared to the PM form. Finally, upon examining Figure 3D, significant differences in Average Power/kg values were identified among the different coffee forms (F = 28.485, *p* < 0.05). The effect size for the average power (Watt/kg) variable, which showed statistically significant differences, was evaluated as “small” (*η*_*p*_^2^ = 0.27). Comparisons among the DK, CK, PC and PM forms indicated that the DK form reached significantly higher Mean Power/kg values compared to all other caffeine forms. The statistically significant differences among the forms are as follows: the DK form is significantly different from the CK, PC and PM forms (*p* < 0.05). Additionally, the CK and PC forms also showed significantly higher Mean Power/kg values compared to the PM form.

## 4. Discussion

The present study aimed to determine the effects of different types of caffeine consumption on cortisol and testosterone levels as well as anaerobic performance in football players. The findings suggest that various forms of coffee have distinct effects on variables such as testosterone, cortisol, lactic acid, and average and maximum power (Watts), which influence athletic performance. There were no significant differences among the DK, CK and PC forms in elevating cortisol levels. However, the PM form showed significantly lower cortisol levels compared to all other forms, which may indicate differences in absorption or physiological response. Conversely, caffeinated coffee was identified as the most effective form in increasing testosterone levels and improving performance metrics. However, the superior performance of caffeinated coffee in enhancing testosterone levels and performance metrics may be due to its more balanced absorption profile. These differences indicate that the absorption rate and bioavailability of caffeine vary depending on the form in which it is consumed, thereby affecting the variables measured in this study to different extents.

The literature indicates that the effects of caffeine can vary from person to person due to variables such as genetic factors and metabolic rate [15]. Additionally, another important factor in the variability of caffeine’s effects is the form in which it is consumed. Caffeine can be found in many forms, such as gels, bars, gums, lozenges, capsules or energy drinks, and these forms vary in absorption level, volume and pH, affecting how quickly caffeine is absorbed from the buccal mucosa and intestines into the bloodstream [16]. The current research findings also support the idea that the bioavailability and absorption rate of caffeine can change depending on the form in which it is consumed. In reviewing the literature, we found that the test-retest reliability and reproducibility of muscle function following acute caffeine intake of 3 mg/kg have been examined. Across three repeated trials, caffeine has been shown to enhance specific measurements of muscle strength, power, and endurance. However, the effects varied from negligible to substantial when individual caffeine trials were compared to placebo. These findings suggest that caffeine can sometimes enhance performance, while at other times it may not, possibly due to session-to-session variability, dosages, and caffeine forms [16,22]. Various studies have shown that the form of caffeine or its method of administration can alter its effects [26,27]. For example, it was found that the absorption time values of three different beverages containing 160 mg of caffeine but with different volumes of solution varied [28]. In another study involving seven participants, it was found that the plasma concentrations of caffeine in capsule form exhibited variability [29]. Additionally, the effects of the temperature and speed of caffeine intake have been investigated, revealing that the temperature and speed of coffee intake can alter both its pharmacokinetic activity and effects [30]. These insights help explain the changes in testosterone, cortisol, lactic acid and anaerobic power performance outcomes observed in our study.

Our study observed that different types of caffeine affected testosterone levels, with the highest testosterone levels being observed in the CK form. Similarly, previous research found that caffeinated gum increased testosterone levels by 53% compared to placebo gum. In terms of sprint performance, the caffeinated group exhibited a 5.8% decrease in mean power output under placebo conditions. However, under caffeine administration, the change in mean power output was only 0.4%, indicating a relative improvement of 5.4% in performance, highlighting the ergogenic effect of caffeine [31]. Stuart et al. reported up to a 5% increase in performance in various rugby-specific training tasks, including sprinting, strength, power, and motor skill-focused tasks, with caffeine administered in capsule or liquid form. This performance enhancement is thought to be partly due to increased testosterone levels resulting from caffeine intake, as higher testosterone levels have been observed to enhance anaerobic performance [31,32,33].

Moreover, the relationship between caffeine and hormone levels appears to vary depending on dosage and pharmacokinetics [14,30]. In this context, the changes in cortisol and testosterone levels observed in our study are consistent with the existing literature. Regarding lactic acid levels, our study’s findings align with other research showing significant results. Mor et al. conducted a study where eight8 trained male athletes were given 25 g of cocoa, 200 mg of caffeine and placebo at different times. After these supplements, participants underwent an Anaerobic Sprint Test, and blood lactate levels were measured post-test. The results showed that caffeine intake resulted in significantly higher blood lactate levels compared to cocoa and placebo [34]. In another study by Barbosa et al., athletes who consumed 300 mg of caffeine before an 800 m run test showed lower levels of lactic acid formation [35]. Similarly, Silveira et al., found that, in nine male participants, caffeine (5 mg/kg) taken at different times led to lower blood lactate levels during a cycling exercise to exhaustion compared to the placebo group [36]. These findings support our study’s results, showing that the CK form had the lowest lactic acid levels.

In addition to changes in lactic acid levels, our study also found significant differences in maximum power and mean power values across different coffee consumption forms, with the highest anaerobic power values observed in the CK form. Lane et al., found that caffeinated gum significantly increased power output compared to placebo in a study examining the effects of caffeine and beetroot juice supplements on performance. The main finding was that power output increased by 3.0% with the caffeine plus beetroot juice combination and by 3.9% with caffeine alone, whereas beetroot juice supplementation alone or in combination with caffeine did not show a significant effect on performance. Specifically, caffeinated gum improved cycling time-trial performance by approximately 3–4% in both men and women, while caffeinated beetroot juice did not demonstrate ergogenic effects [37]. Similarly, Akça et al., examined the effects of different doses (6 mg or 9 mg/kg) of caffeine on 2000 m rowing ergometer performance in male rowers and found that the caffeine groups had better test times and higher mean power outputs compared to the placebo (500 mg glucose) group. Another study on performance cyclists found that caffeinated gum resulted in significant differences in mean power output compared to placebo. The main finding was that the average percentage decrease in power output during the third and fourth exercise sets was 0.4% with caffeinated gum compared to 5.8% with placebo, resulting in a 5.4% performance increase in favor of caffeine [38]. Doherty and Smith conducted a meta-analysis showing that caffeine improved anaerobic performance by more than 12% compared to placebo [11]. Our study results are consistent with these findings in the literature. The increase in testosterone, cortisol, lactic acid and anaerobic power performance due to caffeine and the varying effects depending on the forms of caffeine can be explained by several mechanisms [39]. Caffeine stimulates the central nervous system by blocking adenosine receptors, allowing muscles to work harder. This increases calcium release in muscle cells, leading to faster and stronger contractions. Additionally, caffeine can elevate testosterone and cortisol levels by increasing the release of adrenocorticotropic hormone (ACTH) [14]. Elevated testosterone levels support muscle strength and growth, while cortisol helps meet the body’s energy needs during exercise by regulating energy production and metabolism [40]. Caffeine increases muscle power output by enhancing motor unit activation in the central nervous system. Methods that provide rapid absorption, such as chewing gum, can deliver quicker effects, thus instantly improving performance. Caffeine also promotes the mobilization of fatty acids, encouraging the use of fat for energy production and preserving glycogen stores [41]. The combination of these mechanisms can explain caffeine’s performance-enhancing effects. Differences in the effects of caffeine consumption forms are thought to result from pharmacokinetic effects, bioavailability, absorption rate, consumption method and the impact of other components consumed with caffeine. Similarly, Guest et al., reported that the optimal timing, dosage and source of caffeine intake could vary and influence its effects [42]. In light of the findings and previous research, this study provides compelling evidence that the form of caffeine consumed plays an important role in modulating its effects on testosterone, cortisol, lactic acid levels and anaerobic performance. The findings are consistent with previous literature highlighting the impact of caffeine’s pharmacokinetics, bioavailability and absorption rate, especially when consumed in coffee, which contains additional bioactive compounds such as polyphenols and chlorogenic acids. These compounds not only enhance the absorption of caffeine but also contribute to metabolic changes that improve energy utilization and performance outcomes during exercise [43,44]. As demonstrated in our results, the CK form outperformed both the PC and PM groups in terms of testosterone levels, lactic acid levels and anaerobic power metrics, likely due to these synergistic effects. The caffeine-induced increase in catecholamines during exercise is likely to have triggered the larger testosterone response observed in our study, as supported by prior research showing a similar response. Even without exercise, caffeine ingestion has been shown to increase testosterone levels, as well as muscle size and recovery in animal models. This provides a plausible explanation for the elevated testosterone levels we found, particularly in the CK group [45,46].

Additionally, the lack of significant differences in cortisol levels among caffeinated forms reinforces the notion that the absorption rate and bioavailability of caffeine are key variables influencing hormonal and performance responses. The lower cortisol levels observed in the PM group compared to the caffeinated forms indicate that isolated caffeine consumption may produce different physiological effects compared to its consumption in coffee form. This variability may also be influenced by individual metabolic factors, as supported by the literature pointing to genetic and metabolic variability in caffeine’s effects. Our study’s findings on lactic acid levels support previous research showing that caffeine consumption, particularly in the CK form, delays lactic acid accumulation and enhances anaerobic performance. This is consistent with studies by Barbosa et al. (2017) and Silveira et al. (2018), which found lower lactic acid levels in athletes following caffeine consumption, further emphasizing caffeine’s role in reducing fatigue and improving high-intensity exercise capacity [35,36]. In light of the current study, it is clear that the form in which caffeine is consumed significantly influences its ergogenic effects, including its ability to increase testosterone levels, modulate cortisol and optimize power output. The integration of bioactive compounds in coffee with caffeine presents a more effective ergogenic aid than isolated caffeine forms, demonstrating the importance of considering not only the dose and timing of caffeine but also the form of its consumption. The differences observed among the various coffee forms in the current study are thought to stem from these factors.

### Strengths, Limitations and Recommendations

The strengths of this study include a comprehensive examination of the effects of different caffeine forms (caffeinated coffee, decaffeinated coffee, powdered caffeine and powdered placebo) on cortisol, testosterone, lactic acid levels and anaerobic performance in athletes. The single-blind, counterbalanced, randomized and crossover methodology allowed for an objective assessment of the specific effects of each caffeine form. However, the study was conducted with only 15 male soccer players, limiting the sample size and diversity, which reduces the generalizability of the results. The exclusion of female athletes, as well as the disregard for genetic factors and external influences, further constrains the validity of the findings. Ethnicity is another factor that may account for the obtained results. It is important to consider the potential for placebo effects resulting from the single-blind design used in this study. In a single-blind study, only the participants are unaware of their group assignment, while the researchers know which participants are receiving which treatment. This could lead to placebo effects, where participants’ expectations or beliefs about the treatment may influence their performance or perceived outcomes. To mitigate this, future research could employ a double-blind design, where both participants and researchers are unaware of group assignments, to reduce the potential impact of placebo effects on the study’s results. Additionally, incorporating a more diverse sample and considering genetic variability in future studies could further enhance the robustness and applicability of the findings. Further studies should include more heterogeneous samples of the examined population and have double-blinded and placebo methodologies. These limitations can be addressed in future research with larger sample sizes and inclusion of different sports disciplines.

## 5. Conclusions

In conclusion, the findings of this study indicate that different forms of caffeine have distinct and significant effects on cortisol, testosterone, lactic acid levels and anaerobic performance in football players. Caffeinated coffee (CK) resulted in the highest testosterone levels, the greatest increases in both mean and maximum power output and the lowest lactic acid levels. In terms of cortisol, no significant differences were found between caffeinated coffee (CK), powdered caffeine (PC) and decaffeinated coffee (DK), while the lowest cortisol levels were observed in the placebo (PM) group. These results suggest that the form of caffeine affects both hormonal levels and power performance differently depending on the form consumed. These results highlight the need for further research on the optimal form and dosage of caffeine consumption, and the importance of considering individual differences. Personalized strategies should be developed, considering the form of caffeine consumption and individual variability, especially including genetic factors.

## Figures and Tables

**Figure 1 nutrients-16-03228-f001:**
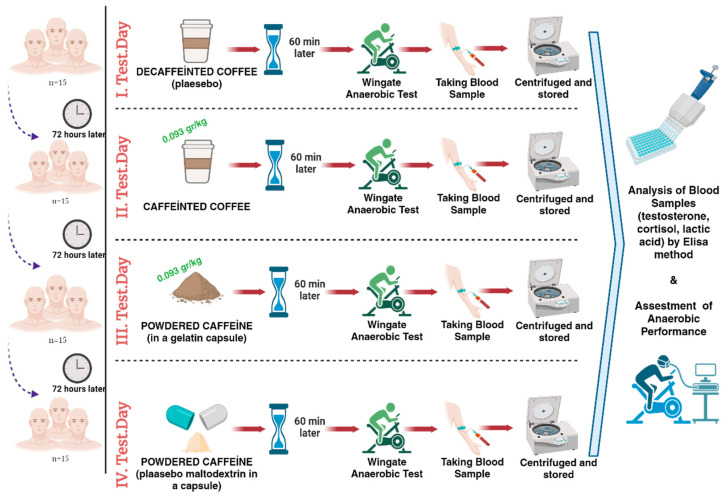
Experimental Design.

**Figure 2 nutrients-16-03228-f002:**
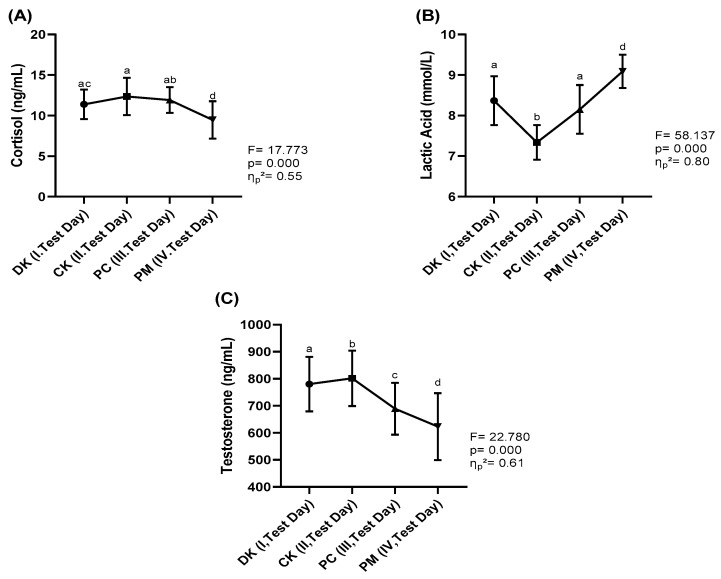
(**A**) Cortisol, (**B**) Lactic Acid and (**C**) Testosterone levels. DK (Decaffeinated Coffee (Placebo), CK (Caffeinated Coffee), PC (Powdered Caffeine in a gelatin capsule), PM (Powdered Caffeine placebo maltodextrin in a capsule). F: ANOVA variance ratio, p: statistical significance value (*p* < 0.05), *η*_*p*_^2^: eta squared effect size. a, b, c, d: There are statistically significant differences between measurements marked with different letters (*p* < 0.05).

**Figure 3 nutrients-16-03228-f003:**
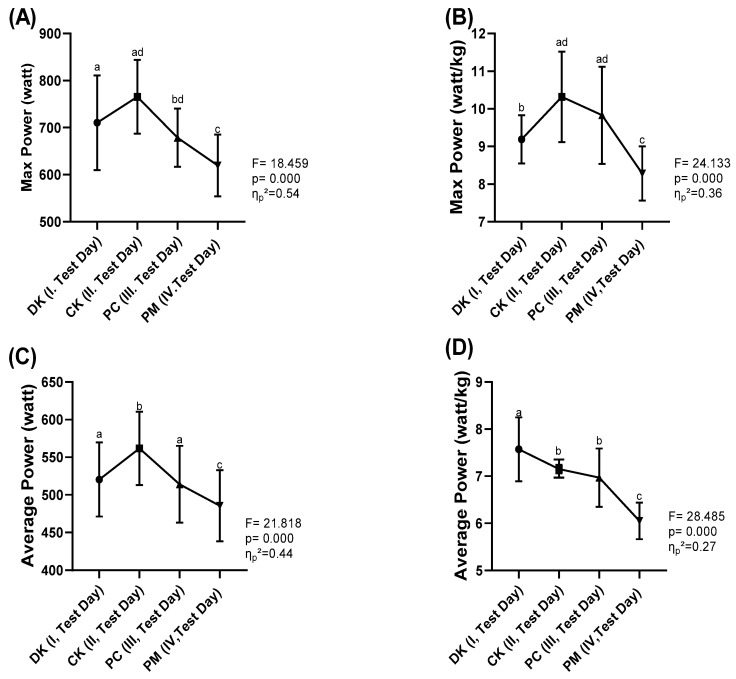
(**A**) Max Power, (**B**) Max Power (Watt/kg), (**C**) Average Power and (**D**) Average Power (Watt/kg). DK (Decaffeinated Coffee (Placebo), CK (Caffeinated Coffee), PC (Powdered Caffeine in a gelatin capsule), PM (Powdered Caffeine placebo maltodextrin in a capsule). F: ANOVA variance ratio, *p*: statistical significance value (*p* < 0.05). *η*_*p*_^2^: eta squared effect size. a, b, c, d: There are statistically significant differences between measurements marked with different letters (*p* < 0.05).

## Data Availability

The data presented in this study are available on request from the corresponding authors. The data are not publicly available due to privacy.

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
