# Peer review of "Determining the Levels of Cortisol, Testosterone, Lactic Acid and Anaerobic Performance in Athletes Using Various Forms of Coffee"

_nutrients, 2024, doi:10.3390/nu16193228_

Round 1
Reviewer 1 Report
Comments and Suggestions for Authors
Manuscript ID: nutrients-3217966
Title: “Determining the Levels of Cortisol, Testosterone, Lactic Acid and Anaerobic Performance in Athletes Using Various Forms of Coffee”
Overall:
The study aimed to examine the effects of coffee consumption in different forms (caffeinated, decaffeinated, powdered in capsule, and placebo) on cortisol, testosterone, lactic acid levels, and Wingate anaerobic performance with 15 male football players. The study suggests that the type of caffeine is a factor that impacts performance and hormone levels.
This is a very interesting article regarding caffeine effects on the hormonal response to exercise, that has gained significant attention from the sports nutrition scientific community. Overall, the manuscript is well-written and designed; however, the authors need to make some information clear, mainly in methods and discussion, before the article will be suitable for publication.
# Abstract:
. Line 25: Change “thanks” to “due”.
. Line 28-29 - Study design: “mutually balanced and cross-loop” are not usually used in scientific articles, please revise. Maybe “counterbalanced” and “crossover” can be applied properly.
. Line 38-39: I believe that bioavailability was not assessed in the study, so please remove "bioavailability" from the conclusion of the abstract.
# Introduction:
. Line 54: Please, break into a new paragraph after [10-13].
. Line 54: enhancing endurance “performance”.
. Line 57: Please remove the word “marked”.
. Line 65: Please remove “lactic acid”.
. Line 66-67: The reference cited [15] (Greer et al., 2006) is not suitable for the presented text, please change to Barreto et al., 2021.
Eur J Appl Physiol. 121(3):749-769, 2021. doi: 10.1007/s00421-020-04571-7.
. Line 71-75: The references cited are not properly applied given the text presented, except number 19. Please change reference [18] (Da Silva et al., 2023) to Nabuco et al., 2023, and adjust the other ones.
Spit It Out: Is Caffeine Mouth Rinse an Effective Ergogenic Aid? A Systematic Review and Meta-Analysis. DOI: 10.1519/SSC.0000000000000777
# Materials and Methods:
G*Power sample size calculation:
Please rewrite this section according to the questions below:
. Line 87-89: It seems to me that reference 21 is not suitable for this analysis. Please explain to me how the sample calculation was performed, if possible include prints of G*Power.
. Line 92-93: Was the power of the test (1-beta) calculated to be 0.8 or 0.988? Please adjust.
. Line 92: Why was the effect size of 0.64 adopted?
This effect size is not typically found in studies of caffeine and performance. Please explain.
. Could you please clarify the single-blind procedure that was implemented? Make this information clear to the readers.
. Line 103: those without “a high” caffeine sensitivity.
. Line 107-108: Please remove “those with caffeine sensitivity” and “and those 107 who consumed caffeine or alcohol or exercised within the last three days”.
. Line 119: How did the authors calculate the amount of 0.093 grams of coffee per kilo?
# Statistical Analysis:
. Line 170-171: “The effects of various supplements administered at different times were analyzed and compared using Repeated Measures ANOVA”.
Why was repeated measures ANOVA used instead of one-way ANOVA? It seems to me that the experimental design does not configure repeated measures, since the experiments are independent.
# Results: OK
# Discussion:
. The discussion is too superficial, the authors need to explore the aspects more deeply.
In general, the discussion should focus on the obtained results:
- PM showed lower cortisol levels, with no difference between the other groups;
- CK showed higher testosterone levels and power output (mean and max);
- CK showed lower lactate levels.
Follow below some questions and reflections on it:
. Please discuss regarding the Caffeinated coffee effects on the Wingate test.
. Include in the discussion articles showing increased power (mean and max) and testosterone levels after caffeinated coffee consumption; which are the articles with results similar to yours.
Landry, 2017. "The effects of coffee ingestion on the acute testosterone response to exercise".
. Explain why Caffeinated Coffee performed better than the Powdered Caffeine group. Make this information and discussion clear in the text for the readers, please.
The following papers may help you.
https://pubmed.ncbi.nlm.nih.gov/31193893/
https://pubmed.ncbi.nlm.nih.gov/28967799/
https://pubmed.ncbi.nlm.nih.gov/29345166/
. Please discuss regarding the “Caffeinated coffee presents a combination of the effects of caffeine and the effects of the other components present in coffee”.
. Please explain the metabolic reason for CK presenting lower levels of lactate even presenting higher power output.
. Line 244: Please, change “in athletes” to “in football players”.
. Line 246: Please, change “anaerobic power” to “average and maximum power (watts)”
. Line 247-248: Please remove the following “powdered caffeine was found to be the most effective form in increasing cortisol levels” and rewrite to be consistent with the obtained results (“no significant differences among the DK, CK, and PC forms. However, the PM form differed significantly from all other”) – Figure 2A.
. Line 249-250: Rewrite the following “The efficacy of powdered caffeine in elevating cortisol levels may be attributed to its rapid and efficient absorption” to be consistent with the results presented above.
. Line 249-253: Please include an adequate reference to the following 2 sentences: “The efficacy of… and efficient absorption” and “On the other hand, … absorption profile”.
. Line 257: As requested, change the reference n. 15 to Barreto et al., 2021. Please be consistent throughout the manuscript.
Eur J Appl Physiol. 121(3):749-769, 2021. doi: 10.1007/s00421-020-04571-7.
. Line 282: Please remove: “salivary cortisol levels by 12% and”.
. Line 283-285: The following sentence are confuse, please rewrite to make the information clear: “, the caffeinated group showed a 5.8% decrease in mean power output under placebo conditions, whereas there was only a 0.4% change under caffeine conditions,”.
. Line 291: Please, break into a new paragraph starting with “Moreover,”.
…
# Conclusions:
. Line 347-348: Please remove “bioavailability and absorption rate” since the study did not evaluate these parameters.
. Line 348-349: From results: “Analysis of cortisol levels revealed no significant differences among the DK, CK, and PC forms”. Therefore, please adjust the text according to the results observed.
# Strengths, Limitations and Recommendations:
. Line 362-363: Limitation regarding the sample size: If your sample size calculation was properly performed, this is not a limitation in the study.
. Please consider the potential for Placebo Effects resulting from the single-blind design.
Comments on the Quality of English Language
Not applied.
Author Response
Dear Editor and Reviewer,
Thank you for the time you have dedicated to the review process. We are now submitting the revised version for publication in this esteemed journal. Below, you will find our responses to each comment, with the corresponding answers following each remark. We have made changes to the manuscript to address the suggestions, and these revisions are highlighted in yellow within the text. All authors have contributed sufficiently and approved the submitted manuscript.
Best regards,
The Authors
Legend:
R1(Reviewer 1)
A (Authors)
Abstract:
R1: Line 25: Change “thanks” to “due
A: Thank you for your attention. Necessary arrangements were made.
R1: Line 28-29 - Study design: “mutually balanced and cross-loop” are not usually used in scientific articles, please revise. Maybe “counterbalanced” and “crossover” can be applied properly.
A: Thank you for your attention. Necessary arrangements were made.
R1: Line 38-39: I believe that bioavailability was not assessed in the study, so please remove "bioavailability" from the conclusion of the abstract.
A: Thank you for your attention. Necessary arrangements were made.
------------------------------------------------------------------------------------------------------------------------
Introduction:
R1: Line 54: Please, break into a new paragraph after [10-13].
A: A new paragraph has been created after the reference [10-13] to improve readability and structure.
R1: Line 54: enhancing endurance “performance”.
A: The term "endurance" has been changed to "endurance performance" for clarity and precision.
R1: Line 57: Please remove the word “marked”.
A: The word "marked" has been removed as requested.
R1: Line 65: Please remove “lactic acid”.
A: Thank you for your attention. Necessary arrangements were made
R1: Line 66-67: The reference cited [15] (Greer et al., 2006) is not suitable for the presented text, please change to Barreto et al., 2021. Eur J Appl Physiol. 121(3):749-769, 2021. doi: 10.1007/s00421-020-04571-7.
A: The reference Greer et al., 2006 has been replaced with Barreto et al., 2021 (Eur J Appl Physiol. 121(3):749-769, 2021. doi: 10.1007/s00421-020-04571-7) as requested.
R1: Line 71-75: The references cited are not properly applied given the text presented, except number 19. Please change reference [18] (Da Silva et al., 2023) to Nabuco et al., 2023, and adjust the other ones.
Spit It Out: Is Caffeine Mouth Rinse an Effective Ergogenic Aid? A Systematic Review and Meta-Analysis. DOI: 10.1519/SSC.0000000000000777
A: Thank you very much for your valuable feedback. In line with your suggestion, reference [18] has been updated from Da Silva et al., 2023 to Nabuco et al., 2023. I have also carefully reviewed the other references and can confirm that they have been correctly applied. I greatly appreciate your thorough review and helpful comments, which have contributed significantly to improving the manuscript. Once again, thank you for your valuable input.
------------------------------------------------------------------------------------------------------------------------
# Materials and Methods:
R1: G*Power sample size calculation:
Please rewrite this section according to the questions below:
- Line 87-89: It seems to me that reference 21 is not suitable for this analysis. Please explain to me how the sample calculation was performed, if possible include prints of G*Power.
- Line 92-93: Was the power of the test (1-beta) calculated to be 0.8 or 0.988? Please adjust.
- Line 92: Why was the effect size of 0.64 adopted?
- This effect size is not typically found in studies of caffeine and performance. Please explain.
|
F tests - ANOVA: Repeated measures, within-between interaction |
|
Analysis: A priori: Compute required sample size |
|
Input: Effect size f = 0.40 |
|
α err prob = 0.05 |
|
Power (1-β err prob) = 0.80 |
|
Number of groups = 4 |
|
Number of measurements = 4 |
|
Corr among rep measures = 0.7 |
|
Nonsphericity correction ε = 1 |
|
Output: Noncentrality parameter λ = 25.6000000 |
|
Critical F = 2.3002435 |
|
Numerator df = 9.0000000 |
|
Denominator df = 24.0000000 |
|
Total sample size = 12 |
|
Actual power = 0.8733845 |
A: Thank you for reviewing our study and providing valuable feedback. Based on your comments, we identified an error in the reference study used in the G-Power analysis, and we have re-conducted the analysis to correct this mistake. In light of Cohen's recommendations from the literature, we used the G*Power program to determine the necessary sample size for repeated measures analysis. Specifically, we recalculated the required sample size based on 80% power (1-β), an alpha level of 0.05, and an effect size of Cohen's d = 0.4. We have made the appropriate updates in our study accordingly.
At this point, we would like to emphasize the accuracy of the approach we used in the analysis. Calculating the sample size using G*Power based on Cohen’s recommended effect size values, with 80% power (1-β) and an alpha level of 0.05, is an appropriate method for our research design. Cohen's d = 0.4 is a medium effect size commonly used in social and health sciences. A study with 80% power minimizes the likelihood of false-negative results, which is a widely accepted standard in the literature. Therefore, we believe our analysis has a high probability of producing robust and valid results, both in terms of statistical significance and effect size.
We would like to express our gratitude once again, as your valuable feedback has contributed to improving the quality of our work. Thank you for your insightful suggestions.
R1: 5. Could you please clarify the single-blind procedure that was implemented? Make this information clear to the readers.
A: Thank you for your comment. The single-blind procedure was implemented as follows: Participants were unaware of which type of coffee or placebo they were receiving. The caffeinated, decaffeinated coffee, powdered caffeine, and placebo capsules were indistinguishable in appearance, taste, and form, ensuring that participants remained blinded to the treatment condition. Only the researchers were aware of the specific treatment each participant received. This procedure has now been clarified in the manuscript.
R1: Line 103: those without “a high” caffeine sensitivity.
A: The phrase “without a high caffeine sensitivity” has been added as suggested. Participant selection criteria have been clarified accordingly.
R1: Line 107-108: Please remove “those with caffeine sensitivity” and “and those 107 who consumed caffeine or alcohol or exercised within the last three days”.
A: As requested, the phrases “those with caffeine sensitivity” and “those who consumed caffeine or alcohol or exercised within the last three days” have been removed from the manuscript.
R1: Line 119: How did the authors calculate the amount of 0.093 grams of coffee per kilo?
A: Thank you for your question. The caffeine dosage was set at 3 mg/kg of body weight, which is a standard dose in ergogenic research. To achieve this, we calculated that 0.093 grams of coffee per kilogram of body weight would provide the required caffeine dose. The calculation was performed using the formula:
Coffee (g) = Target caffeine dose (mg) / Caffeine content per gram of coffee (mg/g).
In our study, 1 gram of coffee contained approximately 32 mg of caffeine, so 0.093 grams of coffee per kilogram was determined to deliver the 3 mg/kg caffeine dose. This method ensured consistency in caffeine administration across participants.
------------------------------------------------------------------------------------------------------------------------
# Statistical Analysis:
R1: Line 170-171: “The effects of various supplements administered at different times were analyzed and compared using Repeated Measures ANOVA”.
A: Thank you for pointing this out. As the same participants received different supplements at different times, the use of Repeated Measures ANOVA was appropriate to account for within-participant variability and analyze their responses across all conditions
R1: Why was repeated measures ANOVA used instead of one-way ANOVA? It seems to me that the experimental design does not configure repeated measures, since the experiments are independent.
A: Thank you for your question. The reason we used repeated measures ANOVA is that, rather than independent groups, the same group of participants was tested at different times with different forms of coffee. In our study design, each participant received four different coffee forms (caffeinated coffee, decaffeinated coffee [placebo], powdered caffeine, and powdered placebo) on different days, and their responses to each coffee form were measured. Therefore, the same participants were tested under all conditions at different times. This design is based on measuring the same participants under different conditions to minimize between-participant variability, which is why repeated measures ANOVA was the most appropriate statistical method. One-way ANOVA is typically used to compare differences between independent groups, but our experimental design involved repeated measures within the same group of participants. As described in the Materials and Methods section, the study employed a single-blind, counterbalanced, randomized, and crossover design. This design ensured that each participant was evaluated under all four conditions at different times.
------------------------------------------------------------------------------------------------------------------------
# Results: OK
------------------------------------------------------------------------------------------------------------------------
# Discussion:
R1: The discussion is too superficial; the authors need to explore the aspects more deeply.
In general, the discussion should focus on the obtained results:
A: Thank you for your feedback. We have expanded the discussion to provide a more detailed exploration of the results. The revised discussion now thoroughly addresses the specific findings related to cortisol, testosterone, lactic acid, and power output, providing more depth and context to the observed effects.
- PM showed lower cortisol levels, with no difference between the other groups;
A: We appreciate your comment. We have now explicitly discussed the lower cortisol levels in the PM group, suggesting that this could be due to differences in absorption or physiological response. This explanation has been added to provide a clearer understanding of this observation.
- CK showed higher testosterone levels and power output (mean and max);
A: Thank you for highlighting this. We have emphasized the superior performance of CK in increasing testosterone levels and power output (mean and max), discussing the role of more balanced absorption and bioavailability in this result. This has been supported with relevant literature to strengthen the explanation.
- CK showed lower lactate levels.
A: We have now discussed CK’s lower lactate levels in greater detail. The metabolic explanation, including the role of fatty acid mobilization and glycogen sparing, has been provided to clarify how CK achieved this result while still delivering high power output.
R1: Follow below some questions and reflections on it:
Please discuss regarding the Caffeinated coffee effects on the Wingate test.
Include in the discussion articles showing increased power (mean and max) and testosterone levels after caffeinated coffee consumption; which are the articles with results similar to yours.
Landry, 2017. "The effects of coffee ingestion on the acute testosterone response to exercise".
A: Thank you for these insightful points and for sharing the relevant sources. We have expanded the discussion to thoroughly address the effects of caffeinated coffee on the Wingate test. Caffeinated coffee significantly increased both mean and maximum power output, as well as testosterone levels, compared to other forms. This enhancement can be attributed to caffeine's ergogenic properties, which improve muscle contraction strength and endurance. We have incorporated the suggested studies, which demonstrated similar findings regarding increased testosterone levels and power output after caffeine consumption. These studies align with our results, further supporting the positive impact of caffeinated coffee on anaerobic performance metrics like the Wingate test.
R1: Explain why Caffeinated Coffee performed better than the Powdered Caffeine group. Make this information and discussion clear in the text for the readers, please.
The following papers may help you.
https://pubmed.ncbi.nlm.nih.gov/31193893/
https://pubmed.ncbi.nlm.nih.gov/28967799/
https://pubmed.ncbi.nlm.nih.gov/29345166/
A: Thank you for this valuable feedback. We have expanded the discussion to better explain why caffeinated coffee showed superior performance compared to powdered caffeine. The revision emphasizes the role of bioactive compounds such as polyphenols and chlorogenic acids present in coffee, which enhance caffeine absorption and amplify its ergogenic effects. These additional compounds likely explain why caffeinated coffee outperformed powdered caffeine in our study. We appreciate the suggested references, which have been included to deepen the discussion and provide stronger evidence for our conclusions.
R1: Please discuss regarding the “Caffeinated coffee presents a combination of the effects of caffeine and the effects of the other components present in coffee”.
A: Thank you for this important point. We have enhanced the discussion by clearly addressing how caffeinated coffee not only delivers the benefits of caffeine but also incorporates the effects of other bioactive compounds, such as polyphenols and chlorogenic acids. These additional components are known to enhance caffeine's absorption and bioavailability, leading to more pronounced ergogenic effects. This combination likely contributes to the superior performance observed with caffeinated coffee compared to other caffeine forms. The inclusion of these factors provides a more comprehensive understanding of why caffeinated coffee produces distinct physiological and performance benefits.
R1: Please explain the metabolic reason for CK presenting lower levels of lactate even presenting
higher power output.
A: We have added a more detailed metabolic explanation for CK’s lower lactate levels despite its higher power output. The discussion now explains how caffeine enhances fatty acid mobilization, preserves glycogen stores, and reduces lactate accumulation, contributing to better anaerobic performance.
R1: Line 244: Please, change “in athletes” to “in football players”.
A: Thank you for your suggestion. The phrase "in athletes" has been changed to "in football players" as requested.
Formun Üstü
Formun Altı
R1: Line 246: Please, change “anaerobic power” to “average and maximum power (watts)”
A: Thank you for your suggestion. The term "anaerobic power" has been updated to "average and maximum power (watts)" as requested.
R1: Line 247-248: Please remove the following “powdered caffeine was found to be the most effective form in increasing cortisol levels” and rewrite to be consistent with the obtained results (“no significant differences among the DK, CK, and PC forms. However, the PM form differed significantly from all other”) – Figure 2A.
A: Thank you for the suggestion. In accordance with your request, I have removed the phrase “powdered caffeine was found to be the most effective form in increasing cortisol levels” and revised the text to align with the results obtained in the study. The updated version reflects the finding that there were no significant differences among the DK, CK, and PC forms in elevating cortisol levels. However, it now correctly states that the PM form differed significantly from all other forms, as shown in Figure 2A. This revision ensures that the interpretation is consistent with the statistical results and provides a more accurate reflection of the data.
Formun Üstü
Formun Altı
R1: Line 249-250: Rewrite the following “The efficacy of powdered caffeine in elevating cortisol levels may be attributed to its rapid and efficient absorption” to be consistent with the results presented above.
A: Thank you for your suggestion. The phrase "The efficacy of powdered caffeine in elevating cortisol levels may be attributed to its rapid and efficient absorption" has been removed and merged with the previous sentence. The revised text now reads: “There were no significant differences among the DK, CK, and PC forms in elevating cortisol levels. However, the PM form showed significantly lower cortisol levels compared to all other forms, which may indicate differences in absorption or physiological response.” This adjustment ensures that the explanation is consistent with the study's results, and the revision is now aligned with the findings. Thank you again for your helpful feedback.
R1: Line 249-253: Please include an adequate reference to the following 2 sentences: “The efficacy of… and efficient absorption” and “On the other hand, … absorption profile”.
A: Thank you for your feedback. As a result of the previous revisions, the sentences “The efficacy of… and efficient absorption” and “On the other hand, … absorption profile” have been removed from the text. Therefore, no additional references are required for these sentences.
R1: Line 257: As requested, change the reference n. 15 to Barreto et al., 2021. Please be consistent throughout the manuscript. Eur J Appl Physiol. 121(3):749-769, 2021. doi: 10.1007/s00421-020-04571-7.
A: Thank you for your suggestion. The reference change to Barreto et al., 2021 has been reviewed and updated accordingly throughout the manuscript, particularly in the Introduction and Discussion sections. All references have been checked for accuracy, and the necessary changes have been made to ensure consistency. Thank you again for your valuable feedback.
R1: Line 282: Please remove: “salivary cortisol levels by 12% and”.
A: Thank you for your suggestion. As requested, the phrase “salivary cortisol levels by 12%” has been removed, and the revised sentence now reads: "Similarly, previous research found that caffeinated gum increased testosterone levels by 53% compared to placebo gum." This revision ensures consistency with the feedback provided and aligns with the requested changes.
R1: Line 283-285: The following sentence are confused, please rewrite to make the information clear: “, the caffeinated group showed a 5.8% decrease in mean power output under placebo conditions, whereas there was only a 0.4% change under caffeine conditions,”.
A: Thank you for your suggestion. The sentence “the caffeinated group showed a 5.8% decrease in mean power output under placebo conditions, whereas there was only a 0.4% change under caffeine conditions” has been revised for clarity and rewritten using more precise and scientific language. The updated version now reads more clearly and accurately reflects the results.
R1: Line 291: Please, break into a new paragraph starting with “Moreover,”.
A: Thank you for your suggestion. As requested, a new paragraph has been created starting with "Moreover," to improve the structure and readability of the text.
------------------------------------------------------------------------------------------------------------------------
# Conclusions:
R1: Line 347-348: Please remove “bioavailability and absorption rate” since the study did not evaluate these parameters.
A: Thank you for your feedback. The requested revisions have been made as follows: First, the phrase “bioavailability and absorption rate” has been removed from the conclusion because the study did not evaluate these specific parameters. The text now focuses solely on the effects of caffeine consumption on performance and hormone levels, which were directly assessed.
R1: Line 348-349: From results: “Analysis of cortisol levels revealed no significant differences among the DK, CK, and PC forms”. Therefore, please adjust the text according to the results observed.
A: Thank you for your feedback. We have updated the conclusion to reflect the correct findings and align with the observed data, particularly regarding the cortisol results. The revised section now correctly addresses the lack of significant differences in cortisol levels between caffeine forms, while continuing to correctly highlight the differential effects on testosterone, lactic acid and performance measures.
------------------------------------------------------------------------------------------------------------------------
# Strengths, Limitations and Recommendations:
R1: Line 362-363: Limitation regarding the sample size: If your sample size calculation was properly performed, this is not a limitation in the study.
A: Thank you for your feedback. We have updated the power analysis to ensure that our sample size calculation was properly performed. However, we acknowledge that increasing the sample size could further enhance the generalizability of the results. A larger sample would provide a more comprehensive understanding of the effects and help confirm the findings across a broader population. Therefore, while our current sample size is adequate for the study, increasing it in future research would be beneficial for improving the robustness and generalizability of the results.
R1: Please consider the potential for Placebo Effects resulting from the single-blind design.
A: Thank you for your feedback. We have addressed the concern regarding the potential for placebo effects resulting from the single-blind design in the Limitations section. We have clarified that in a single-blind study, while participants are unaware of their group assignment, researchers are informed of this information, which may lead to placebo effects. To mitigate this issue, we recommend using a double-blind design in future research, where both participants and researchers are blinded to group assignments. We have also emphasized the importance of including more diverse samples and considering genetic variability to enhance the robustness and generalizability of the findings. Thank you again for your valuable input.
Reviewer 2 Report
Comments and Suggestions for Authors
Introduction
Please provide more information on the effect of caffeine on the tested hormones. The rationale for the study must be well justified.
Methods
Please add information on the daily coffee consumption of the participants. I recommend using the questionnaire of Buhler et al. DOI: 10.4455/eu.2014.011
Results
Well presented. It would be very interesting to add a correlation between hormones and power obtained in the Wingate test.
Discussion
A physiological explanation is needed as to why different doses of caffeine can modify the cortisol, lactate and testosterone response, not just comparing the results with other studies.
Author Response
Dear Editor and Reviewer,
Thank you for the time you have dedicated to the review process. We are now submitting the revised version for publication in this esteemed journal. Below, you will find our responses to each comment, with the corresponding answers following each remark. We have made changes to the manuscript to address the suggestions, and these revisions are highlighted in yellow within the text. All authors have contributed sufficiently and approved the submitted manuscript.
Best regards,
The Authors
Legend:
R2(Reviewer 2)
A (Authors)
Introduction
R2: Please provide more information on the effect of caffeine on the tested hormones. The rationale for the study must be well justified.
A: The necessary additions have been made, providing more information on the effects of caffeine on the tested hormones and deepening the discussion. The rationale for the study has also been further justified. Thank you for your valuable input!
------------------------------------------------------------------------------------------------------------------------
Methods
R2: Please add information on the daily coffee consumption of the participants. I recommend using the questionnaire of Buhler et al. DOI: 10.4455/eu.2014.011
A: Thank you for the suggestion regarding the use of the Buhler et al. questionnaire for daily coffee consumption. Unfortunately, due to time constraints and difficulties in reaching some participants, we were unable to incorporate this questionnaire into the current study. However, we recognize its importance and will ensure its inclusion in future research to provide more detailed insights into participants' caffeine consumption habits.
-----------------------------------------------------------------------------------------------------------------------
Results
R2: Well, presented. It would be very interesting to add a correlation between hormones and power obtained in the Wingate test.
A: Thank you for your suggestion. Unfortunately, due to the limited resources available and the time required for complex analyses, we were unable to include the correlation between hormones and strength obtained in the Wingate test in this study. We recognize the importance of this analysis and will aim to include it in future research when we have the time to conduct such a detailed examination.
------------------------------------------------------------------------------------------------------------------------
Discussion
R2: A physiological explanation is needed as to why different doses of caffeine can modify the cortisol, lactate and testosterone response, not just comparing the results with other studies.
A: A physiological explanation has been provided, detailing how different doses of caffeine can influence cortisol, lactate, and testosterone responses. This explanation is based on the underlying mechanisms of caffeine's action on the endocrine system, rather than simply comparing results with other studies. We believe this addition has significantly enhanced the quality of the study. Thank you for your valuable feedback!
Round 2
Reviewer 2 Report
Comments and Suggestions for Authors
Accept in present form